# Participatory Assessment of Potato Production Systems and Cultivar Development in Rwanda

**Jean Baptiste Muhinyuza [1],\*, Alphonsine Mukamuhirwa [1], Marie Chantal Mutimawurugo [1], Jean Damascène Mazimpaka [1], Delitha Girumugisha Muhinyuza [2] and Rodomiro Octavio Ortiz Rios [3]**

1   Department of Crop Science, College of Agriculture, Animal Science and Veterinary Medicine, University of Rwanda, Musanze P.O. Box 210, Rwanda
2   Department of Economics, College of Business and Economics, University of Rwanda, Kigali P.O. Box 4285, Rwanda
3   Genetics and Plant Breeding, Swedish University of Agricultural Science, SE-23422 Lomma, Sweden
\*   Correspondence: mujohnbapt25@gmail.com

**Abstract:** Potato cultivars grown in Rwanda are very old, low yielding and not amenable to food processing. High yielding and late blight tolerant cultivars for this country should be evaluated at different agro-ecozones prior to releasing them to farmers, who are yet to be integrated into potato breeding. The objectives of this study were to assess farmers' preferred traits in potato cultivars and to gather knowledge from farmers about potato clones bred in Rwanda. Four respondents per village in 36 villages each for the districts of Musanze, Burera and Nyamagabe participated in the survey, whose questionnaire was about farm size, gender balance, land allocated to potatoes and other main crops, potato "seed" sourcing, potato production constraints and most important potato attributes. Potato was rated as the most important food and cash crop. 'Kirundo', 'Cruza', 'Mabondo' and 'Victoria' were the most popular cultivars. Among them, Mabondo' was the most resistant to the oomycete *Phytophthora infestans* causing late blight. Potato production in Rwanda is limited by lack of improved cultivars, high temperature, drought, acidic soil, pathogens, insects, weeds, inadequate storage of tubers as planting material, post-harvest technology, low market price of tubers at harvest, lack of access to credit, climate change, and gaps such as inadequate fertilizer and fungicide applications. The most important cultivar attributes were high tuber yield, host plant resistance and high specific gravity or dry matter.

**Keywords:** farmers' preferred attributes; participatory assessment; Rwanda

## 1. Introduction

Potato (*Solanum tuberosum* L.; $2n = 4x = 48$ chromosomes) is among the four major food crops grown worldwide [1]. In terms of human consumption and total grown area, potato ranks third after wheat (*Triticum aestivum* L.) and rice (*Oryza sativa* L.) [2,3]. Potato is first in total production among root and tuber crops (*Dioscorea batatas* Deene) [1]. The 100 potato growing states are located in the tropics and subtropics [4]. The total estimated area under potato production in the world is 17 584 000 hectares with a total production of 366 298 000 tons (Table 1) [5]. Asia is the largest potato producer with around 43% of global production, followed by Europe (38%), America (13%) and Africa (6%) (Figure 1) [5]. In Africa, Algeria is the leading potato producer, followed by Egypt, Malawi and South Africa, while Rwanda is the fifth-largest producer (Table 2). In Eastern and Central Africa (ECA), potato is an important food security crop [6]. Potatoes grow well in several parts of Rwanda but their production is concentrated at high altitudes (mainly above elevations of 1800 m above sea level), and two to three crops can be grown in a year. Small family farms grow potato along with beans (*Phaseolus vulgaris* L.) and maize (*Zea mays* L.), and harvest on average almost 10 tons per hectare [7].

**Table 1.** Potato area, production and yield by region (2021).

| Region | Harvested Area (×1000 ha) | Production (×1000 T) | Tuber Yield (T ha$^{-1}$) |
|---|---|---|---|
| Africa | 1,828 | 25,026 | 13.7 |
| Asia | 9,325 | 184,937 | 19.8 |
| Europe | 4,823 | 109,783 | 22.8 |
| Latin America | 1,051 | 20,161 | 19.2 |
| North America | 5,550 | 26,390 | 47.9 |
| World | 17,584 | 366,298 | 20.8 |

Source: [5].

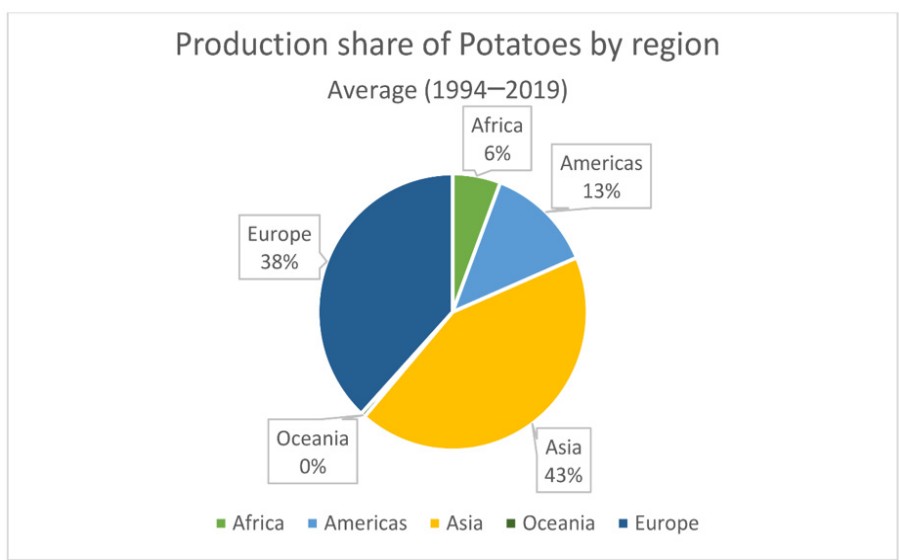

**Figure 1.** Potato production by region (%). Source: [5].

**Table 2.** Six largest potato-producing African states (2021).

| Country | Tuber Harvest (T) |
|---|---|
| Algeria | 4,928,028 |
| Egypt | 4,800,000 |
| Malawi | 4,535,955 |
| South Africa | 2,252,000 |
| Rwanda | 2,240,715 |
| Kenya | 2,192,885 |

Source: [5].

Potato provides various nutrients, and it may be an important source for enhanced nutrition to the growing population worldwide [2]. Its content of carbohydrates are high and it produces considerable energy with large amounts of protein and vitamins, especially vitamin C, as well as minerals such as P, Ca, Zn, K and Fe [5]. Such important nutritional value makes potato an efficient crop in combating malnutrition [5]. Based on its overall economic importance, the volume of potatoes produced and consumed worldwide has increased substantially [1,2].

Potato is, after banana, the second-most-important food crop of Rwanda (Table 3), where it also has an important role as a cash crop [5]. There are more than 70,000 potato farmers, grouped in 30 cooperatives, which harvest over 19,000 tons monthly towards the end of the cropping season [8]. 'Kinigi', 'Kirundo', 'Mabondo', 'Cruza', 'Sangema' are the most popular grown cultivars, among others (Table 4), in Rwanda. Recently, four new cultivars, namely, 'Nkunganire' (CIP 39280.84), 'Twihaze' (CIP 393371.58), 'Izihirwe' (CIP 396018.14) and 'Ndeze' (CIP 393280.84) from the International Potato Center (CIP) were released after field testing across sites and over years in the country [8]. Moreover, five

locally bred cultivars ('Twigire', 'Seka', 'Cyerekezo', 'Ndamira', 'Gisubizo' and 'Jyambere' were included in the national performance trial (NPT) prior to their official release in 2020 [9]. The Rwanda Agriculture Board (RAB) potato sub-program began a local potato breeding program, with support from the Alliance for Green Revolution in Africa (AGRA) and CIP. Currently, RAB has advanced six elite potato breeding clones to the pre-release stage (Table 5) due to the following attributes: high yield and host plant resistance to the oomycte *Phytophthora infestans* causing late blight, which is the most devastating disease in the potato production system of Rwanda. This crossbreeding endeavor aims to evaluate farmers' preferred potato attributes and to collect new genotypes for further evaluation in the different agricultural ecologies of Rwanda.

**Table 3.** Production of five major food crops in Rwanda.

| Commodity | Tuber Harvest (T) |
| --- | --- |
| Plantain and banana | 2,749,150 |
| Potato | 1,789,400 |
| Cassava | 1,377,210 |
| Dry bean | 840,072 |
| Sweet potatoes | 327,497 |

**Table 4.** List of potato cultivars released in Rwanda after breeding by CIP and ISAR.

| Cultivar | Year of Release | First Year NARS Testing | CIP No. (or Other Source) | Pedigree |
| --- | --- | --- | --- | --- |
| 'Atzimba' | 1980 | 1972 | 720,045 | 'Leona' × PEN3PD-23 |
| 'Montzama' | 1980 | 1972 | 720,049 | 'Furore' × (US129.2' × 'Katahdin') |
| 'Sangema' | 1980 | 1972 | 800,949 | Mutation of 'Rosita' |
| 'Muhabura' | 1980 | 1976 | | |
| 'Bufumbira' | 1980 | 1976 | | |
| 'Gahinga' | 1984 | 1980 | 720,097 | Furore × Greta |
| 'Gasore' | 1984 | 1980 | 800,955 | 'Gracilia' × 'Soraya' |
| 'Nseko' | 1984 | 1980 | 720,055 | 54-Q-2 × 'Amarilla de Puebla' |
| 'Petero' | 1984 | 1980 | | |
| 'Kinigi' | 1984 | 1980 | 378,699.2 | 'Puca Toralapa' × YY-1 |
| 'Cruza 148' | 1985 | 1981 | 720,118 | 'Montserrate' × unknown |
| 'Mabondo' | 1988 | 1982 | 800,983 | 720097 × 378676.6 |
| 'Kirundo' | 1989 | 1982 | | |
| 'Turbo' | 1989 | 1982 | (SM 80-13) | SM 69-17 × VE 74-45 |
| 'Obelix' | 1989 | 1982 | (ZPC 77 L 55) | 'Ostara' × 'Renska' |
| 'Ngunda' | 1992 | 1988 | 381,395.1 | 378,493.915 × Bulk Mexico |
| 'Mizero' | 1992 | 1987 | 386,003.2 | BL-2.9 × R128-6 |
| 'Gikungu' | 1992 | 1988 | 387,233.24 | 382,124.6 × India 1039 |
| 'Mugogo' | 1992 | 1988 | 383,140.6 | 37,493.738 × Bulk Plaisted |
| 'Nderera' | 1992 | 1984 | 381,381.3 | 378,493.915 × Bulk Precoz |
| 'Kigega´' | 1992 | 1988 | 383,120.14 | VHF-69.1 × Bulk Mexico |
| 'Victoria´' | 1989 | 1996 | 381,381.2 | 378,493.15 × Bulk Precoz |

Source: Labarta [10].

**Table 5.** List of six elite potato clones tested since 2012 that are in the pre-release stage.

| No. | Parentage | Code | Tuber Yield (T/ha) |
| --- | --- | --- | --- |
| 1 | 'Gikungu' × 22M83 | RWPOT012.8 | 17.65 |
| 2 | 'Gikungu' × 22M88 | RWPOT012.12 | 21.12 |
| 3 | 'Gikung' × 3M73 | RWPOT012.16 | 24.84 |
| 4 | 'Kinigi' × 22M85 | RWPOT012.28 | 20.54 |
| 5 | 'Kinigi' × Nderera M10 | RWPOT012.29 | 24.40 |
| 6 | 'Nderera' × 22M34 | RWPOT012.31 | 21.02 |

## 2. Materials and Methods

The highlands (>1800 m above sea level) of northwest Rwanda were mostly the target areas for this research along with Buberuka Highland region and the Nile-Congo Crest in the southwest. The chosen districts were Musanze in the highland of volcanic soils, Burera in the Buberuka area, and Nyamagabe within the Congo-Nile divide, which are main potato-growing sites in the country due to their soil fertility, land productivity, and suitable climates for this tuber crop (Figure 2), [9]. These sites have a bimodal rainfall pattern, i.e., short and long rains during October to mid-December and March to June, respectively [11]. Nonetheless, rainfall occurs almost always in these sites, thus allowing potato planting throughout the year. Their average annual temperature and rainfall are at 16 °C and 1500 mm, respectively [11]. The main food crops cultivated in the selected districts are potato, maize, beans, wheat, peas, sorghum and vegetables.

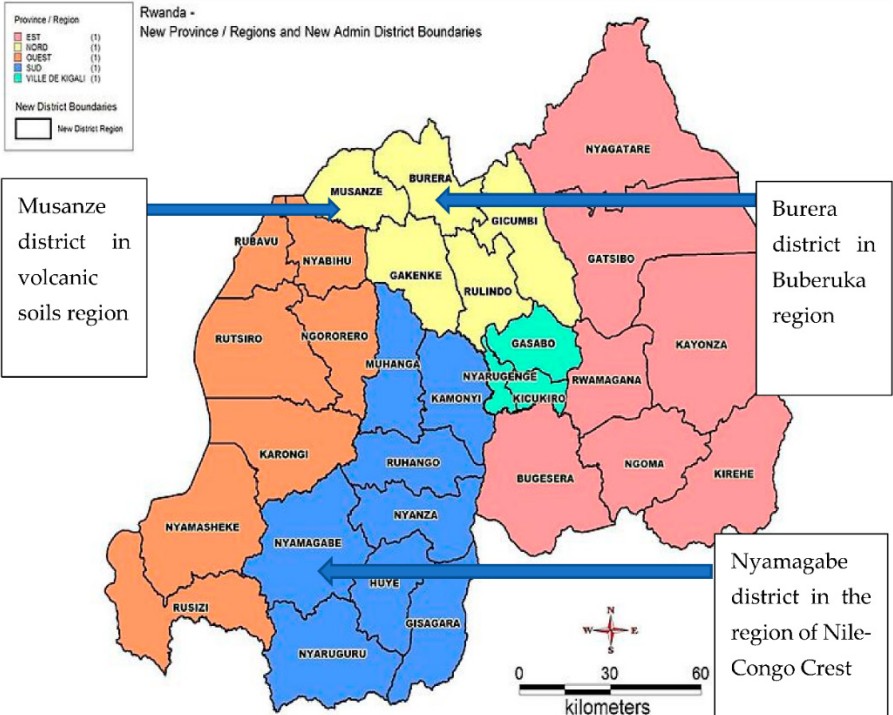

**Figure 2.** Administrative map of Rwanda.

The highland areas were selected following a purposive sampling procedure while considering their importance in potato production in the country [7]. We used random sampling to select villages and farmers therein with the assistance of the village leaders and extension workers. Hence, the survey involved 36 villages and 144 respondents in three districts and was conducted from February to April 2022. A questionnaire was used for farmers to provide information regarding their farm size, land allocated to potato and other crops, and source of potato-planting material. Data from the structured survey questionnaire along with secondary data from previous surveys and other reports were analyzed using descriptive statistics.

## 3. Results

Potato was grown throughout the year as a food and cash crop. Table 6 displays the time period that potatoes were grown and the number of times they were planted per year. Most farmers grew potato at least twice a year. Musanze was the first district that adopted potato, and thereafter were Nyamagabe and Burera. The number of respondents interviewed and the gender composition are presented in Table 7.

**Table 6.** Potato production by district included in the survey.

| District | Number of Years Growing Potato | Number of Times per Year Growing Potato | Size of the Largest Plot Growing with Potato (ha) |
|---|---|---|---|
| | Mean | Mean | Mean |
| Musanze | 17 | 3 | 2 |
| Burera | 11 | 2 | 2 |
| Nyamagabe | 12 | 2 | 2 |

**Table 7.** The number of farmers interviewed and gender composition (percentage in brackets).

| District | Male | Female | Total |
|---|---|---|---|
| Musanze | 27 (71.0) | 11 (29.0) | 38 (26.4) |
| Burera | 32 (57.1) | 24 (42.8) | 56 (38.9) |
| Nyamagabe | 28 (56.0) | 22 (44.0) | 50 (34.7) |

*Land allocation*: The land area for growing potatoes was larger than that used for other food crops. The average household land size was 1 ha, and 0.9 ha of this (90%) was used for farming (Table 8), of which 41.5 to 56.3% was for potato production (Table 9).

**Table 8.** Average farm size and cultivated land per household.

| District | Total Farm Size (ha) | Total Cultivated Land (ha) |
|---|---|---|
| | Mean | Mean |
| Musanze | 0.5 | 0.3 |
| Burera | 1.0 | 0.6 |
| Nyamagabe | 1.4 | 1.0 |

**Table 9.** Importance of crops grown and their land per household.

| Crop | Income Generation | | Family Food Use | |
|---|---|---|---|---|
| | Percentage | Mean Area (ha) | Percentage | Mean Area (ha) |
| Potato | 56.5 | 0.5 | 41.5 | 0.5 |
| Maize | 13.6 | 0.2 | 23.1 | 0.2 |
| Vegetables | 2.2 | 0.1 | 2.7 | 0.4 |
| Peas | 3.9 | 0.3 | 4.3 | 0.2 |
| Beans | 11.5 | 0.2 | 18.8 | 0.2 |
| Wheat | 9.7 | 0.3 | 5.1 | 0.3 |
| Sweet potato | 2.2 | 0.1 | 3.9 | 0.2 |
| Bananas | 0.1 | 0.3 | 0.3 | 0.3 |
| Sorghum | 0.3 | 0.2 | 0.3 | 0.2 |

*Tuber planting material sources and use of production inputs*: The sources of potato tuber planting material in the study areas are presented in Table 10. Traders (39.6%) and open markets (36.1%) were the main sources for farmers to purchase their potato tubers for planting. Both research institutions and private enterprises had a minor role as planting materials providers (11.1% and 9.7%, respectively). There were few farmers (3.5%) who kept their own planting materials from their own potato tuber harvests.

**Table 10.** Source of potato tuber planting material.

| Tuber Planting Sources | Number of Farmers | Percentage |
|---|---|---|
| Own field | 5 | 3.5 |
| *Trader* | *57* | *39.6* |
| Open market | 52 | 36.1 |
| Private company | 14 | 9.7 |
| Research institution | 16 | 11.1 |

*Major production constraints:* The participatory assessment revealed that the most important potato production constraints were the lack of access to credits for the growing season (mean score: 43.3), lack of suitable high yielding potato cultivars (42.3), insufficient tubers for use as clean planting materials (40.3) and cultivar susceptibility to *P. infestans* (39) (Table 11).

**Table 11.** Potato production constraints in three districts of Rwanda.

| | District | | | | | | | |
|---|---|---|---|---|---|---|---|---|
| **Constraint** | **Musanze (*N* = 48)** | | **Burera (*N* = 48)** | | **Nyamagabe (*N* = 48)** | | | |
| | **Mean** | **Rank** | **Mean** | **Rank** | **Mean** | **Rank** | **Overall Mean** | **Overall Rank** |
| Late blight susceptibility | 45 | 3 | 27 | 5 | 45 | 3 | 39 | 4 |
| Unclean tuber planting material | 43 | 4 | 36 | 2 | 42 | 4 | 40.3 | 3 |
| Poor storage facilities | 26 | 7 | 35 | 3 | 29 | 6 | 30 | 6 |
| Dormancy period | 32 | 6 | 16 | 8 | 28 | 7 | 25.3 | 7 |
| Low yield | 46 | 2 | 34 | 4 | 47 | 1 | 42.3 | 2 |
| Low price | 37 | 5 | 26 | 6 | 32 | 5 | 31.7 | 5 |
| Lack of fertilizers | 9 | 9 | 9 | 9 | 13 | 9 | 10.3 | 9 |
| Lack of pesticides | 8 | 10 | 9 | 10 | 12 | 10 | 9.7 | 10 |
| inaccessibility to Credit | 47 | 1 | 37 | 1 | 46 | 2 | 43.3 | 1 |
| Soil degradation | 12 | 8 | 20 | 7 | 23 | 8 | 18.3 | 8 |
| Mean | 30.5 | | 24.9 | | 31.7 | | 29.02 | |

*N*: Number of farmers per district that participated in the survey.

*Importance of pests*: Table 12 lists the most important potato pests as determined by farmers, who grouped them, with the assistance of the interviewer, into four categories: fungal diseases (mostly late blight), bacterial wilt, viruses, and insects. Late blight was the main pest in Musanze, while bacterial wilt was the most important pest affecting the crop in Burera. Bacterial wilt was the least important in Nyamagabe (Table 12). *P. infestans* caused significant crop damage (25–50%), whereas bacterial wilt and virus infections affected 26.5% and 26.6% of the crops, respectively (Table 13). Serious crop damage (i.e., >50%) was mostly due to bacterial wilt (31%) and late blight (18.2%) (Table 13).

**Farmers' preferred cultivars:** Kirundo', 'Cruza', 'Mabondo', 'Victoria', 'Gikungu' and 'Sangema' were the cultivars grown across the three districts included in the survey (Table 14). 'Kirundo' (mean score = 2.7), 'Cruza' (1.9), 'Mabondo' (1.8) and 'Victoria' (1.2) were the most important cultivars, but they did vary in their ranking across sites. 'Rutuku' was the most important cultivar grown in Burera, but this cultivar is a name given to all the cultivars with red skin. It could be 'Kinigi', 'Victoria' or 'Gikungu', which are the most important red cultivars available therein.



**Table 12.** Major potato pests in three districts of Rwanda (Score: 0–5).

| Pests | District | | | | | | Overall Mean | Overall Rank |
|---|---|---|---|---|---|---|---|---|
| | Musanze | | Burera | | Nyamagabe | | | |
| | Mean | Rank | Mean | Rank | Mean | Rank | | |
| Late blight | 4.0 | 1 | 2.0 | 2 | 3.0 | 1 | 3.0 | 1 |
| Bacterial wilt | 3.0 | 2 | 3.0 | 1 | 1.0 | 4 | 2.3 | 2 |
| Viral diseases | 1.0 | 3 | 1.0 | 3 | 2.0 | 2 | 1.3 | 3 |
| Insect pests | 0.0 | 4 | 0.0 | 4 | 1.0 | 3 | 0.3 | 4 |
| Overall Mean | 2.0 | | 1.5 | | 1.5 | | | |

**Table 13.** Crop damage levels (%) due to potato pathogens across three districts of Rwanda.

| Type of Damage | Late Blight | Bacterial Wilt | Viruses |
|---|---|---|---|
| Complete crop loss | 9.1 | 5.6 | 5.7 |
| Serious damage (+ 50%) | 18.2 | 31.0 | 14.7 |
| Important damage (25–50%) | 29.8 | 26.5 | 26.6 |
| Non-important damage (< 25%) | 29.6 | 24.3 | 36.4 |
| Without damage | 13.3 | 12.6 | 16.6 |
| Total | 100.0 | 100.0 | 100.0 |

**Table 14.** Potato cultivars grown in Rwanda (Score: 0–5).

| Cultivar | District | | | | | | Average | Ranking |
|---|---|---|---|---|---|---|---|---|
| | Musanze | | Burera | | Nyamagabe | | | |
| | Mean | Rank | Mean | Rank | Mean | Rank | | |
| 'Cruza' | 1.4 | 7 | 1.2 | 4 | 3.2 | 1 | 1.9 | 5 |
| 'Mabondo' | 2.7 | 4 | 2.4 | 3 | 0.5 | 7 | 1.8 | 6 |
| 'Makoroni' | 1.1 | 9 | 1.1 | 6 | - | - | 1.1 | 9 |
| 'Kirundo' | 3.3 | 3 | 3.7 | 2 | 1.2 | 4 | 2.7 | 4 |
| 'Victoria' | 1.0 | 10 | 1.2 | 5 | 1.3 | 3 | 1.2 | 12 |
| 'Gikungu' | 0.5 | 14 | 1.0 | 7 | 0.4 | 8 | 0.6 | 17 |
| 'Sangema' | 0.0 | 15 | 0.4 | 10 | 0.7 | 5 | 0.4 | 20 |
| 'Petero' | 4.0 | 2 | - | - | - | - | 4.0 | 3 |
| 'Kinigi' | 4.1 | 1 | - | - | - | - | 4.1 | 2 |
| 'Nyirakabondo' | 0.9 | 12 | - | - | - | - | 0.9 | 14 |
| 'Nyabizi' | 1.7 | 6 | - | - | - | - | 1.7 | 7 |
| 'Bineza' | 1.0 | 11 | - | - | - | - | 1.0 | 13 |
| 'IPP' | 0.8 | 13 | - | - | - | - | 0.8 | 16 |
| 'Kigega' | 1.7 | 5 | - | - | 0.2 | 9 | 0.9 | 11 |
| 'Rwishaki' | 1.3 | 8 | - | - | - | - | 1.2 | 10 |
| 'Rutuku' | - | - | 4.6 | 1 | - | - | 4.6 | 1 |
| 'Mbumbe' | - | - | 0.4 | 12 | - | - | 0.4 | 22 |
| 'Nderera' | - | - | 0.4 | 11 | - | - | 0.4 | 21 |
| 'Mizero' | - | - | 0.7 | 8 | - | - | 0.7 | 15 |
| 'Makerere' | - | - | 0.6 | 9 | - | - | 0.6 | 18 |
| 'Gasore' | - | - | - | - | 0.1 | 10 | 0.1 | 23 |
| 'Nyirangeli' | - | - | - | - | 0.0 | 11 | 0.0 | 24 |
| Local | - | - | - | - | 1.4 | 2 | 1.4 | 8 |
| 'Mugogo' | - | - | - | - | 0.5 | 6 | 0.5 | 19 |
| 'Kenya' | - | - | - | - | 0.0 | 12 | 0.0 | 25 |

*Late blight susceptibility in cultivars*: Table 15 provides the ranking of cultivars according to their susceptibility to *P. infestans*. 'Cruza' was the least susceptible cultivar (mean score = 2.9), followed by 'Mabondo' (2.5) and 'Kirundo' (1.7) in the highland regions.

'Kinigi' and 'Rutuku' were, however, the least susceptible cultivars in Musanze and Burera, respectively, while 'Cruza' was the least susceptible in Nyamagabe.

**Table 15.** Cultivar susceptibility to late blight in three leading potato-producing districts of Rwanda (Score: 0–5).

| | District | | | | | | | |
|---|---|---|---|---|---|---|---|---|
| | Musanze | | Burera | | Nyamagabe | | | |
| Cultivar | Mean | Rank | Mean | Rank | Mean | Rank | Average | Rank |
| Mabondo | 3.3 | 2 | 2.6 | 3 | 1.6 | 3 | 2.5 | 4 |
| Cruza | 2.1 | 6 | 4.4 | 1 | 2.2 | 1 | 2.9 | 3 |
| Kirundo | 2.6 | 4 | 2.0 | 4 | 0.5 | 8 | 1.7 | 8 |
| Victoria | 1.0 | 11 | 0.0 | 10 | 0.8 | 5 | 0.6 | 16 |
| Sangema | 0.0 | 15 | 0.3 | 8 | 0.9 | 4 | 0.4 | 17 |
| Gikungu | 0.6 | 12 | 1.0 | 7 | 0.6 | 7 | 0.7 | 13 |
| Kigega | 1.1 | 9 | - | - | 0.3 | 9 | 0.7 | 14 |
| Makoroni | 2.9 | 3 | 1.9 | 5 | - | - | 2.4 | 5 |
| Kinigi | 4.5 | 1 | - | - | - | - | 4.5 | 1 |
| Petero | 2.1 | 5 | - | - | - | - | 2.1 | 6 |
| Nyirakabondo | 0.9 | 10 | - | - | - | - | 0.9 | 11 |
| Bineza | 1.9 | 7 | - | - | - | - | 1.9 | 7 |
| Nyabizi | 0.2 | 14 | - | - | - | - | 0.2 | 19 |
| IPP | 0.3 | 13 | - | - | | | 0.3 | 18 |
| Rwishaki | 1.3 | 8 | - | - | - | - | 1.3 | 10 |
| Rutuku | - | - | 4.2 | 2 | - | - | 4.2 | 2 |
| Makerere | - | - | 0.0 | 11 | - | - | 0.0 | 23 |
| Nderera | - | - | 0.8 | 6 | - | - | 0.8 | 12 |
| Mizero | - | - | 0.1 | 9 | - | - | 0.1 | 21 |
| Gasore | - | - | | | 0.6 | 6 | 0.6 | 15 |
| Mugogo | - | - | - | - | 0.0 | 12 | 0.0 | 24 |
| Nyirangeli | - | - | - | - | 0.2 | 10 | 0.2 | 20 |
| Local | - | - | - | - | 1.8 | 2 | 1.6 | 9 |
| Kenya | - | - | - | - | 0.1 | 11 | 0.1 | 22 |

*Farmers' preferred potato traits:* The most important attributes sought by farmers are high tuber yield (mean score = 4.2), host plant resistance to pathogens and pests (3.6), and high specific gravity/dry matter content (3.4), though it could vary among districts (Table 16). Other important features of potato cultivars that farmers liked were early maturity (1.9) and dormancy period (1.1) (Table 16), as well as marketability, tolerance to poor soils, and big tubers with round shape (Table 17).

**Table 16.** Ranking of farmers' preferred potato characteristics (Score: 0–5).

| | District | | | | | | | |
|---|---|---|---|---|---|---|---|---|
| | Musanze | | Burera | | Nyamagabe | | Overall | |
| Characteristic | Mean | Rank | Mean | Rank | Mean | Rank | Mean | Rank |
| High yield | 4.2 | 1 | 4.4 | 1 | 4.1 | 1 | 4.2 | 1 |
| Host plant resistance | 3.6 | 3 | 3.4 | 3 | 3.5 | 2 | 3.5 | 2 |
| Good taste | 0.4 | 6 | 0.8 | 6 | 0.5 | 6 | 0.6 | 6 |
| Short dormancy | 1.1 | 5 | 0.9 | 5 | 1.4 | 5 | 1.1 | 5 |
| Early maturity | 1.9 | 4 | 2.0 | 4 | 2.0 | 4 | 1.9 | 4 |
| High dry matter content | 3.8 | 2 | 3.7 | 2 | 2.7 | 3 | 3.4 | 3 |

**Table 17.** Most grown potato cultivars with their advantages and disadvantages as rated by farmers.

| District | Cultivars | Advantages | Disadvantages |
|---|---|---|---|
| Musanze, Burera and Nyamagabe | 'Kinigi', 'Kirundo', 'Rutuku' and 'Mabondo' | High tuber yield and dry matter content, marketability, host plant resistance to late blight, big tuber size and round tuber shape | Susceptibility to bacterial wilt |
| Musanze | 'Petero' | High tuber yield and dry matter content | Susceptibility to pathogens and pests |
| Musanze, Burera and Nyamagabe | 'Cruza' | High tuber yield, host plant resistance to pathogens, tolerance to poor acidic soil | Low dry matter content, small-to medium tuber size, late maturity |
| Nyamagabe | Local | Host plant resistance to pathogens and pests | Late maturity, small tuber size, low yield |
| Musanze, Burera and Nyamagabe | 'Victoria' | High tuber yield, big tuber with round shape, early maturity | Susceptible to pathogens, low dry matter content |

## 4. Discussion

Our research shows that potato is a main food security crop and source of cash income for rural households in Rwanda. Maize, beans, wheat, peas, vegetables and sorghum are other main food crops grown along with potato in this country, which remains one of the most densely populated worldwide. Landholdings are very small, with 1 ha as the average land size per farmer, who use more than 50% of it for producing potato. The main sources of tuber planting materials for the potato are traders and open markets. Farmers seldom had access to clean tuber planting materials, thus increasing the incidence and severity of important pathogens and pests affecting potato farming in Rwanda. Infected tuber planting materials spread pathogens and pests for this vegetatively propagated tuber crop. Selection for host plant resistance in newly bred cultivar and clean planting materials will likely lower both the incidence and severity of important pathogens and pests affecting potato in Rwanda.

Lacking access to credits for purchasing inputs during the cropping season or not having suitable high tuber yielding planting materials, as well as insufficient clean planting materials, the available cultigen germplasm susceptibility to *P. infestans*, the tuber dormancy period, low market prices, soil degradation, inaccessibility to fertilizers and pesticides were cited by farmers as their major limitations for achieving big potato harvests in Rwanda, where late blight remains the most important pest, as previously noted [11,12].

'Kirundo', 'Cruza', 'Mabondo', 'Victoria', 'Gikungu' and 'Sangema' are the leading cultivars grown in Rwanda, but some of them are susceptible to *P. infestans*. The cultivar 'Mabondo' is the least susceptible cultivar across the districts included in the survey, whereas 'Victoria' is the most susceptible, as also noted after many years of field testing by ISAR [11].

Potato farmers in Rwanda seek cultivars showing high and stable tuber yield, host plant resistance to pathogens (particularly *P. infestans*) and pests, and with high dry matter/specific gravity. Likewise, early maturity, short dormancy, marketability, tolerance to poor soils, large round tubers are also appreciated by farmers in this country.

## 5. Conclusions

The pro-active participation of farmers in potato breeding should be pursued because their engagement is important for selecting suitable breeding clones in cultivar development and ensuring the successful adoption of newly bred germplasm. Hence, farmers should be involved in the development of potato cultivars suiting their preferences.

**Author Contributions:** J.B.M. led the design of the study with co-authors' inputs. He performed data recording and analysis, as well as led the writing of the manuscript. The other co-authors: D.G.M. and J.D.M. participated in data collection, and curation, A.M. and M.C.M. participated in interpreting the data analysis, while R.O.O.R. provided text and edits to the evolving manuscript drafts. All authors have read and agreed to the published version of the manuscript.

**Funding:** This research was partially funded by the University of Rwanda.

**Data Availability Statement:** Not applicable.

**Acknowledgments:** The authors are grateful to University of Rwanda for the funding provided for the research from which the manuscript ensued. They also thank to the who took part in the study.

**Conflicts of Interest:** The authors declare no conflict of interest. The funder had no role in the design of the study; in the collection, analyses, or interpretation of data; in the writing of the manuscript; or in the decision to publish the results.

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
