# Peer review of "Participatory Assessment of Potato Production Systems and Cultivar Development in Rwanda"

_sustainability, doi:10.3390/su142416703_

Round 1
Reviewer 1 Report
The author analyzed the problems existing in the local potato production system through a large amount of survey and research data, which has important guidance for future industrial policy formulation. The authors are to be commended for undertaking such work. However, there are concerns with the current version of the manuscript.
1.The sample survey distribution map is necessary because it reflects the representativeness of the samples.
2. The authors should introduce the content of the questionnaire design, the education level of the interviewees, the time span of the survey and other specific details, and so on.
3. The data analysis in Materials and methods part is too simple. How to analyze the description and statistics of large samples? How are important statistics handled?
4. The conclusion part should be revised thoroughly and concisely. The results need to be summarized and then draw a conclusion. There are too many descriptive sentences about the results in this section.
Author Response
All my responses, comments and corrections were included in the revised manuscript.

Reviewer 2 Report
Please see the attachment.

Author Response
All my responses, comments and answers were included in the revised manuscript for your consideration. the manuscript was send in the form of track change

Round 2
Reviewer 1 Report
Accept in present form.